# Detecting Syntactic Change with Pre-trained Transformer Models

**Liwen Hou**
Harvard University
liwenhou@fas.harvard.edu

**David A. Smith**
Northeastern University
dasmith@ccs.neu.edu

## Abstract

We investigate the ability of Transformer-based language models to find syntactic differences between the English of the early 1800s and that of the late 1900s. First, we show that a fine-tuned BERT model can distinguish between text from these two periods using syntactic information only; to show this, we employ a strategy to hide semantic information from the text. Second, we make further use of fine-tuned BERT models to identify specific instances of syntactic change and specific words for which a new part of speech was introduced. To do this, we employ a part-of-speech (POS) tagger and use it to train corpora-specific taggers based only on BERT representations pretrained on different corpora. Notably, our methods of identifying specific candidates for syntactic change avoid using any automatic POS tagger on old text, where its performance may be unreliable; instead, our methods only use untagged old text together with tagged modern text. We examine samples and distributional properties of the model output to validate automatically identified cases of syntactic change. Finally, we use our techniques to confirm the historical rise of the progressive construction, a known example of syntactic change.

## 1 Introduction

Many researchers have taken advantage of the availability of pre-trained language models to study semantic change (e.g., Giulianelli et al., 2020; Hu et al., 2019; Fonteyn, 2020), but computational studies of syntactic change are more sporadic, relying on specially trained models (Merrill et al., 2019). In this paper, we investigate the extent to which BERT (Devlin et al., 2019) is able to distinguish between the syntax of English from the early 1800s and that of the late 1900s. To do so, we use the British Hansard corpus, which consists of a digitized version of the U.K. Parliamentary debates starting in the early 1800s and spanning

across two centuries. We isolate all of the parliamentary debates from the 1830s and the 1990s.[1] These debates can be found on the U.K. Parliament Website's Hansard archive[2].

We fine-tune *BERT-base-uncased*, which has around 110 million parameters and was pre-trained on approximately 3.3 billion words. When training or testing BERT on a sentence, we first tokenize it using the original *BERT-base-uncased* tokenizer, as implemented by Hugging Face (Wolf et al., 2020).

We focus on two questions: First, to what extent can BERT distinguish text from different time periods on the basis of syntactic features alone? Second, can we use BERT to help identify words or sentences whose analysis reflects syntactic change in an unsupervised way?

In §2, we find evidence that BERT can be used to detect syntactic differences between the English of the 1830s and that of the 1990s. Specifically, even after we control for confounders, a fine-tuned BERT model can still distinguish between these two periods even when the semantic information is hidden.

In §3, we describe how we train classifiers that can assign POS tags to words based on a 1830s and a 1990s understanding of English, respectively. Our main contribution comes in §4, where we use these classifiers to identify words whose analysis reflects syntactic change. Our method can automatically find words whose part of speech has changed. Moreover, our method avoids running a POS tagger on old text, since automatic tagging may be unreliable in that scenario. We emphasize that our change-detection techniques work with no manual annotation and identify examples of language change without any manual filtering; see Appendix B for a list of the top automatically detected words whose part of speech has changed.

---

[1] We chose these decades because they were the earliest and latest decade with large amounts of data in every year.

[2] https://www.hansard-archive.parliament.uk/

To the best of our knowledge, such unsupervised syntactic change detection has not been previously done.

Finally, we confirm a known change (an increase in the frequency of progressives) using our classifiers. We show that a BERT model fine-tuned on text from the 1830s displays a poorer understanding of the progressive construction compared to a similar model trained on text from the 1990s.[3]

## 2 Detecting Syntactic Change While Masking Semantics

In this section, we check the extent to which BERT is sensitive to changes in the syntax of English over time. This acts as a sanity check for later sections, in which we use BERT in a different way to identify specific instances of syntactic change.

We start by fine-tuning *BERT-base-uncased* on the classification task of predicting whether sentences come from the 1830s or the 1990s. To do so, we use the representation of the CLS token in the last layer of BERT and add a linear layer to it; this model is then fine-tuned using labeled sentences from the 1830s and the 1990s.

After we fine-tune it on 100,000 sentences per time period, the BERT model achieves 98.6% accuracy; however, it may use semantic information to distinguish between text from the two time periods, and we want to limit ourselves to syntactic information only in order to check the extent to which BERT can detect syntactic change over time.

To make BERT use syntactic information only, we mask semantic information by replacing all the open class words in each sentence with their POS tags, obtained using Stanford CoreNLP (Manning et al., 2014). In order to leverage BERT's pretrained knowledge of English, we then replace each POS tag with a common English word that usually has that tag (in a similar vein to Gulordava et al., 2018); this way, the input to BERT looks more like "colorless green ideas" and not "JJ JJ NNS".

After replacing words (other than forms of "to be" and "to have", prepositions, and determiners) with POS tags and after replacing each POS tag with a representative word for that part of speech, we fine-tune BERT (with the added layer described above) on the resulting data. The BERT model performs surprisingly well on this classification task, achieving over 90% accuracy for the task of predicting whether a sentence comes from the 1830s or the 1990s. However, there are still several ways for such a model to use non-syntactic information. First, the model may use sentence lengths, since the distribution of the sentence lengths is different for the 1830s than for the 1990s. Second, the BERT model may simply be using the counts of the POS tags rather than real syntactic information.

To mitigate the length confounder, we place restrictions on the lengths of our training sentences. We run experiments separately for the length bins 11-15 words, 21-25 words, 31-35 words, and 20 words. To check the extent to which BERT relies on syntactic information other than the counts of tags, we separately train a model on shuffled input data, in which the sequence of tags in the sentence appears out of order. Such shuffled data retains no information other than the tag counts; we then compare whether BERT trained on unshuffled data can outperform BERT trained on shuffled data. The results are summarized in Table 1.

We compare these results to a logistic regression model that takes in the frequencies for the 100 most common vocabulary items in the replaced corpus[4] (plus a catch-all category for the remaining items) and the sentence length as features. Such a logistic regression model achieves an accuracy of 85.45%, which means it performs about as well as BERT fine-tuned on shuffled tags but loses to BERT when the latter is fine-tuned on unshuffled tags. Using sentence length as the only feature yields an accuracy of 60.33%.

The conclusion of this experiment is that the fine-tuned BERT model must be using syntactic information to tell apart sentences from the 1830s and the 1990s. Indeed, even when the length of the sentence is fixed, and even when the words have been replaced by their parts of speech, the model gains advantage from seeing the true order of the tags as they appeared in the sentence, instead of seeing shuffled tags. This advantage must fundamentally come from differences in syntax that are present between the 1830s and 1990s sentences (and moreover, these differences must take a form that is not simply tag counts).

Having established that BERT is sensitive to changes in the syntax of English, we next train BERT in a different way to identify specific instances of such change. Importantly, from here on

---

[3]The code for our experiments is available here: https://github.com/houliwen/detecting-syntactic-change

[4]In addition to the typical words that represent different parts of speech, this replaced corpus also contains prepositions, determiners, and forms of "to be" and "to have".

| sent. length | shuffled replaced | unshuffled replaced | shuffled plain | unshuffled plain |
|---|---|---|---|---|
| 11-15 | 83.37 | 87.74 | 94.70 | 96.69 |
| 21-25 | 86.44 | 90.54 | 97.39 | 98.63 |
| 31-35 | 89.37 | 93.12 | 98.20 | 98.90 |
| 20 | 85.20 | 89.30 | 96.60 | 97.40 |

Table 1: Accuracy of BERT fine-tuned on different types of data for predicting time period. All models were trained on 20,000 sentences and tested on 2,000 (half from the 1830s and half from the 1990s); the exception was the last line, for which only 10,000 training sentences were used due to lack of data. Since test data contained an even mixture of 1830s and 1990s sentences, baseline accuracy is 50%. These numbers are based on a single run; however, note that an accuracy of $p$ will have standard deviation around $\sqrt{p(1-p)/2000}$, which ranges from $0.23\%$ to $0.83\%$ for the above, with higher accuracies having lower standard deviations.

out, we avoid using a POS tagger on older English.

## 3 Training Classifiers

We use BERT to identify specific instances of change. To do so, we train two POS taggers, one based on text from the 1830s and the other based on text from the 1990s; in the subsequent sections, we will use these taggers and their errors to identify words or sentences that display syntactic change. In this section, we will focus on describing how these taggers are trained and their performance. Crucially, we will train these taggers without ever tagging text from the 1830s; all POS tags are only ever associated with text from the 1990s.

We start by fine-tuning two instances of *BERT-base-uncased*: one on text from the 1990s and the other on text from the 1830s. This is done via Masked-Language Modeling in PyTorch (Paszke et al., 2019), where we process one sentence at a time with 15% of the tokens masked, and the model is trained to predict the masked tokens. We use each fine-tuned model to obtain BERT token representations for 1990s text and then average the token representations for all the WordPiece tokens within each word to obtain a representation of a word. We do this twice: once with the model fine-tuned on text from the 1830s and once with the model fine-tuned on text from the 1990s (yielding two different embeddings of each word). However, even though we have models trained on two time periods, we use both models only to embed words from the 1990s and not any words from the 1830s. At the end of this process, each word from the 1990s has two different BERT embeddings.

Next, using POS tags obtained from Stanford CoreNLP for the 1990s portion of the British Hansard, we train a 2-layer perceptron (with one hidden layer of size 50) [5] to assign a Penn Treebank tag[6] to each BERT representation of each word. An input to this 2-layer perceptron is a BERT embedding of a word (extracted as described above), and the output is a POS tag. This classification task ignores punctuation and has 21 classes: We merge the 'NN', 'NNP', 'NNS', and 'NNPS' tags into one single tag called 'Noun', after which we take the 20 most common tags ('Noun', 'IN', 'DT', 'JJ', 'VB', 'PRP', 'RB', 'VBN', 'VBZ', 'VBP', 'CC', 'MD', 'TO', 'VBD', 'CD', 'VBG', 'PRP$', 'WDT', 'WP', 'WRB') and then group the remaining POS tags into a catch-all class called 'other'. We actually train two such perceptrons separately: one for the 1830s BERT embeddings (of words from the 1990s only) and one for the 1990s BERT embeddings (also only of words from the 1990s).

Note that the training and test data for this procedure consist only of sentences from the 1990s; this is done because the POS tags obtained from Stanford CoreNLP are likely to be more accurate on our modern data. Using the representations from the final layer of BERT (when the classifier has access to 50,000 training sentences, containing 815,578 words, and when the test set has 50,000 sentences, containing 816,428 words), the accuracy obtained by the 1990s model is 97.6%, and the accuracy of the 1830s model is 97.2%.

We repeat the above analysis using POS tags from Trankit (Nguyen et al., 2021) instead of Stanford CoreNLP because Trankit is a state-of-the-art parser and tagger.[7] By using Trankit tags to train and test the above classifiers based on BERT representations, the accuracy obtained by the 1990s model was 98.4% (instead of 97.6% using CoreNLP tags) and the accuracy obtained by the 1830s model was 98.0% (instead of 97.2% using CoreNLP tags). The higher accuracy means that the task became easier when predicting Trankit tags instead of CoreNLP tags; since Trankit tags are also more accurate, it suggests that our pipeline is working as expected.

In §4 and later, we will use these pipelines (and the differences between their predictions) to iden-

---

[5]We tried hidden layer sizes 100 and 200; this hyperparameter did not make a large difference.

[6]www.cis.upenn.edu/~bies/manuals/tagguide.pdf

[7]https://trankit.readthedocs.io/en/latest/performance.html

tify specific instances of syntactic change between the 1830s and the 1990s.

# 4 Identifying Words that Underwent Change

We now attempt to use the POS taggers trained above to identify words that underwent change. We employ two methods to do this: First, we find the instances for which the two trained models disagree on a POS tag (indicating that the model familiar with text from the 1830s would tag the word differently compared to the one familiar with text from the 1990s). The second method involves masking out the target word to see if the 1830s model predicts the tag correctly on the masked word. If the masked word is tagged correctly but the unmasked word is tagged incorrectly, it suggests that the 1830s model can correctly parse the rest of the sentence but is specifically confused by the target word; when this happens, it indicates that the target word may have undergone change. This latter strategy is depicted in Figure 1.

## 4.1 Tagging Disagreements

After training, we have two new POS taggers, one based on an 1830s understanding of English and one based on a 1990s understanding. We now run these taggers on test data (from the 1990s) to find words on which they disagree. In particular, we find words for which the model trained on text from the 1990s agrees with the Trankit tag but where the 1830s model disagrees. The cases where such a disagreement occurs should be the cases in which an 1830s grammar leads the model to incorrectly parse the sentence (likely due to a change in English grammar for the relevant word or perhaps one of its neighbors). Moreover, we use our classifiers to extract probabilistic predictions for the tags, and we search for word instances where probability assigned to the correct tag by the 1830s model was at least 50 percentage points lower than that assigned by the 1990s model.

We do this for each word in each sentence of the 1990s test data. To help isolate interesting cases, we filter the words under consideration to those that occur at least 10 times in the 1830s text (indicating that they are not new words). We remove numbers, punctuation, and words written entirely in capital letters (likely acronyms). We also restrict to words for which such a tagging disagreement occurred at least twice. We then sort the resulting words

by the total number of tag disagreements for that word, normalized by the frequency of the word in the 1990s data, with additional weight given to words which were common in the 1830s (to reduce the number of rare words in the final list). We run this experiment on 100,000 test sentences, using classifiers trained on 25,000 sentences each.

We then manually examine the top resulting words to check whether the identified words have truly undergone change; see §4.3. One example can be seen in the following sentence: "My right hon. Friend last met M. Chevenement to discuss co-operative weapons development among other topics on 4 May." In this example, the model trained on text from the 1990s tags the word "co-operative" as an adjective whereas the model trained on the 1830s believes that "co-operative" is a noun; the 1990s model is correct in this case.

## 4.2 Masking and Confusion

As before, we use both the 1830s model and the 1990s model to predict the POS tag of each word in held-out 1990s sentences. This time, we additionally repeat this process with one word in the sentence masked out: we ask both the 1990s model and the 1830s model to predict the POS tag of the masked word (without knowing what it is). We mask out each word in each sentence, one at a time; since the BERT embedding for a single word depends on the entire sentence, this requires passing the same sentence repeatedly through BERT with a different word masked each time. To speed up this process, we skip applying the mask to words that are too common (within the one thousand most common words in the 1990s), since these constitute the majority of word occurrences and yet are less likely to exhibit syntactic change.

After this process, each word in each sentence (except those we have deliberately skipped) will have four different contextual BERT embeddings: one by the 1830s model when the word was masked, one by the 1830s model when the word was unmasked, one by the 1990s model when the word was masked, and one by the 1990s model when the word was unmasked.

We now examine words for which the 1830s model gives a better prediction of the POS tag when the word is masked compared to when it is unmasked. When this happens, it means that the 1830s model correctly identified which tag to expect in the position of the masked word, and yet

There is no example of sovereign countries which have given up their currencies.

There is no example of [MASK] countries which have given up their currencies.

BERT (fine-tuned on 1830s or 1990s)

( )  →  2-layer perceptron

**POS tag prediction**

Trankit tag: JJ
1990s masked: JJ
1990s unmasked: JJ >90%
1830s masked: JJ  44%
1830s unmasked: Noun 80%

Figure 1: A depiction of the models' predictive process. The 1830s model predicts the correct tag when the word "sovereign" was masked (the tags assigned the highest likelihood after JJ were VBG with 21% and DT with 20% probability) but predicts it incorrectly when it was unmasked; this is the opposite of the usual pattern, in which the prediction is worse in the masked scenario. This type of tagging error indicates that the 1830s model expects "sovereign" to be a noun instead of an adjective.

after the mask is lifted (and the word itself was revealed) the 1830s model "changed its mind" and assigned the wrong tag instead. In such cases, we say that the 1830s model was "confused" by the token. We restrict to cases where the 1990s model correctly tagged the word, even when it was unmasked; this helps remove Trankit tagging errors as well as typographical errors in the text.

We then consider the remaining words for which the 1830s BERT model was confused more than once; specifically, we restrict to words for which there were at least two separate occurrences where the 1990s model tagged the word correctly (with a confidence of at least 90%) and the 1830s model assigned a probability that is at least 25 percentage points higher to the correct POS tag when seeing the masked version of the sentence than it did when seeing the unmasked version. We run this experiment on 100,000 test sentences; to train the 1990s and 1830s models, we use 25,000 sentences each (just as in the previous section). To help isolate interesting cases, we employ the same filters and the same sorting strategy as we did in §4.1.

An example of a word that we found this way is "material", which was an adjective in the early 1800s but whose noun usage is modern. For instance, in the sentence "the inquiry will be free to publish whatever material it wishes," Trankit correctly tags "material" as a noun, and so does the 1990s pipeline, but the 1830s pipeline only gives the correct tag when the word "material" is masked out in that sentence; when it is unmasked, an adjective is predicted.

A list of the top resulting words (with exam-

ple sentences and tags provided by the models) is given in Appendix B. The words in this table were automatically generated with no manual curation.

### 4.3 Evaluation

To evaluate the resulting words, we manually analyzed the top 20 candidates produced by §4.1 and the top 20 candidates produced by §4.2. For each candidate word, we examined 10 randomly sampled sentences containing that word from the 1990s and also 10 sentences from the 1830s, and we manually checked whether their syntax appeared to have changed.

Of the top 20 words from §4.1, seven (or 35%) were identified as having undergone syntactic change, with a part of speech in the 1990s that is either new or greatly increased in frequency. Of the top 20 words from §4.2, 11 (or 55%) were identified as having undergone syntactic change.

We note also that these two approaches identified different sets of words: Of the top 20 words on each list, there were only 4 words of overlap, while of the top 100 there were 15 words of overlap.

We also attempt an automatic evaluation by using Trankit to tag words from the 1830s and checking if these tags changed. Of course, our identification methods purposely avoided using a tagger on old text, since it may be unreliable, and a manual check reveals that the tagger made mistakes. Still, as a sanity check, we use it to compare the two methods for identifying syntactic change.

For the purpose of automatic evaluation, we say that a word has changed and gained a new POS if the tag is at least twice as likely to appear for

1990s usages of the word than for 1830s usages of the word. Under this criterion for change, and using the Trankit tags for both 1830s and 1990s data, we get that the top 100 candidates from the simpler identification method (§4.1) contained 20 changed words, while the top 100 candidates from our masking strategy (§4.2) had 35 changed words. Note that of the manually evaluated words, the automatic evaluation agreed with 75% of the human judgements. These should be interpreted with caution, but these results agree with our conclusion that the masking strategy is better at identifying changed words.

## 5 Confusion Scores at the Sentence Level

In this next experiment, we try to discern the extent to which masking out a word in a sentence makes the sentence more difficult for a fine-tuned BERT model to parse. The idea is that, for a typical word, once the word is masked out the model has less information about the sentence, and its syntactic understanding should therefore decrease; on the other hand, for a word that has an atypical or new usage, the model may end up being more confused after seeing the word than it is when the word is masked out. In the latter situation, we would expect the model's syntactic understanding of the sentence to improve if the word is masked out because, from the model's point of view, the word is misleading. A similar idea was used in §4.2 at the word level; in this section we will look at the accuracy of the tags for the whole sentence before and after a single word is masked.

For example, consider the following sentence: "Opposition Members think that is how peace processes work." A model trained on English from the 1830s is likely to think that the word "work" is a noun rather than a verb. This could lead such a model to consider an incorrect way to parse this sentence, where the subject "peace" is doing the action of processing the object "work". In this case, if the word "work" is masked out, the sentence will be more likely to be parsed correctly.

As a proxy for the success of the parsing of the sentence, we feed in the sentence to the classifiers from §3 to get predicted tags for all the words in the sentence. We can use these classifiers to get probabilistic predictions, and in particular, we can get a probability assigned to the correct tag for each word in the sentence (using Trankit tags as ground truth). We then multiply together all

these probabilities to get a measure of the overall probability the model assigns to the correct tags of the sentence. We can repeat this for both the 1830s and 1990s classifiers, and we can also repeat this when a specific word is either masked or unmasked.

We say a model is "confused" if the product of probabilities assigned to the correct (Trankit) tags is higher when the word is masked than when it is unmasked. For instance, in the above example sentence "Opposition Members think that is how peace processes work," the 1830s model may assign a wrong tag to the word "processes" when the word "work" is unmasked, but it might correctly parse the sentence if the word "work" is masked (in the latter case, it will assume the masked word is an unknown verb, while in the former case it will try to parse the sentence with a noun in that position if it believes that "work" is a noun). In other words, if the 1830s model is confused by a word, it could be because this word underwent change from the 1830s to the 1990s. We create a "confusion score" by taking the logarithm of the ratio of the two aforementioned products (i.e. the product of probabilities assigned to the other words when the selected word is masked, and the corresponding product when the selected word is unmasked).

We can get such a confusion score for any word in any sentence, and for either the 1830s model or the 1990s model. We focus on sentences from the 1990s that were not part of the training data. Since calculating confusion scores is resource-intensive, we focus on a pre-selected set of word instances, such as those identified as likely having undergone syntactic change in §4.

### 5.1 Comparing Selected Words to Random Words

We select 10,000 word occurrences for which the tag is predicted differently by the 1830s model and the 1990s model (as discussed in §4.1). For each word occurrence, we calculate a confusion score. We then compare these scores to the confusion scores of 10,000 randomly chosen word instances. Our goal is to check whether the confusion scores of the 10,000 selected words were higher than those of random words.

One possible confounder is that the selected words may be rarer than the random words. To adjust for this, we choose the random words more carefully: for each non-random word in the first set, we pick a random word conditional on its overall

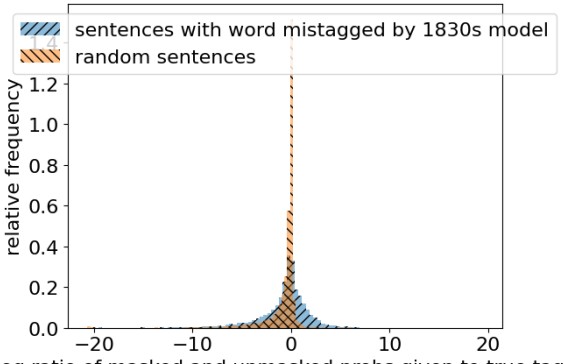

Figure 2: Distribution of confusion scores for selected words and random words.

frequency being close to that of the original word. We define close to be within 10 percentage points in the percentile of their frequencies. The resulting distribution of the confusion scores for the random words had lighter tails than the distribution of the selected words, especially on the right-hand side of the distribution. See Figure 2.

As the figure shows, the selected word instances (for which the 1990s model and 1830s model disagree on the tags) have a higher proportion of high confusion scores than random words. Therefore, calculating confusion scores for the selected word occurrences is a more resource-efficient way to find instances of high confusion scores.

## 5.2 Further Refinements

In addition to calculating confusion scores for a word in a given sentence using the 1830s model, we also calculate confusion scores in a similar way using the 1990s model. Recall that the confusion score tells us the extent to which the BERT-based tagger performs better when a word is masked out than when it is not. Since we have two such taggers (one fine-tuned on parliamentary speeches from the 1830s and one fine-tuned on speeches from the 1990s), we can get two confusion scores for a single word in a single sentence.

In order to zoom in on syntactic change, we restrict our attention to instances where the 1990s confusion score is less than or equal to zero. Since the confusion score is the logarithm of the ratio of masked to unmasked probabilities, this means we are restricting to word instances where the 1990s model gave a better prediction of the tags when the word was unmasked than it did when the word was masked. Applying this restriction helps to remove cases that are confusing for reasons other

than language change: for example, sentences that contain typographical errors, sentences that are incorrectly parsed by Trankit, sentences that were ungrammatical when originally spoken, or other special cases. By focusing on instances that the 1830s model found confusing but that the 1990s model did not find confusing, we increase the likelihood of finding syntactic change.

## 5.3 Manual Evaluation

We generate a list of top 100 candidates for syntactic change based on these confusion scores. Of those 100 sentences, our manual evaluation determined that 33 sentences were valid instances of syntactic change. We also computed the sum of the reciprocal ranks of the 33 manually selected sentences that exhibited syntactic change. This sum was 2.802, while the maximum possible sum of reciprocal ranks would be $\frac{1}{1} + \frac{1}{2} + \ldots + \frac{1}{33} \approx 4.089$.

## 6 Detecting the Rise of the Progressive

In this section, we check the extent to which our techniques can identify syntactic change that was previously known to exist.

The frequency of the progressive has been increasing over time (e.g. Leech et al., 2009; Hou and Smith, 2021). An example of a progressive is "he is reading" (as opposed to "he reads"); the former has a form of "to be" followed by a verb that ends in "-ing". We identify progressive constructions using the dependency parse obtained from Trankit (Nguyen et al., 2021). We then mask out the present participle if it has the tag "VBG" and if the previous word is an inflection of the verb "to be". (This means that we mask out the tokens constituting the "-ing" word as well as the "-ing" suffix itself.)

Next, we use both the 1990s model and the 1830s model to predict the POS tags of all the words in these masked sentences. We compare these predictions to each other and also to the predictions that both models give for the unmasked versions of these sentences. Finally, we take randomly chosen words in randomly chosen sentences as a control group and repeat the process for this group.

We summarize the results in two figures. First, Figure 3 shows the confusion scores for the 1830s model on the progressive sentences compared to random sentences. As the figure shows, the 1830s model is negatively confused when we mask out progressives, which means that it gets the predic-

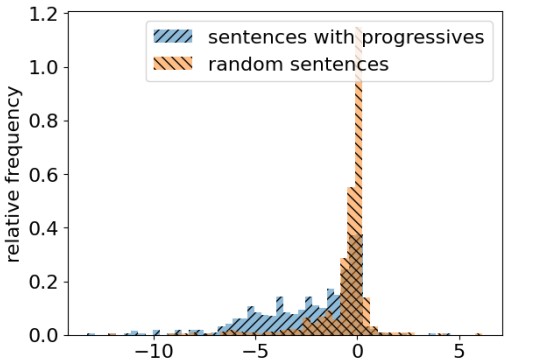

Figure 3: A histogram for the confusion scores of the 1830s model on progressives and on random sentences.

tion right more often on the unmasked version of these sentences than it does on the masked versions. This makes sense, as it corresponds to the insight that the 1830s model has a difficult time correctly tagging the masked progressives.

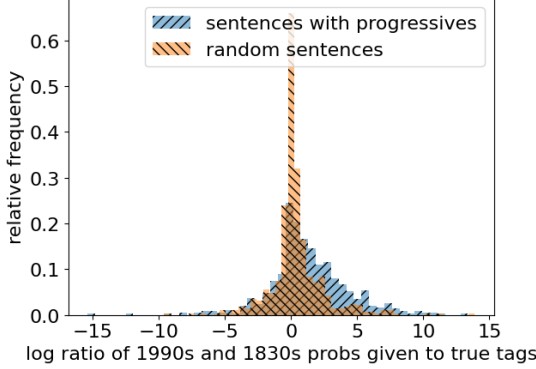

Figure 4: A histogram for the logarithm of the ratios between the products of probabilities assigned to the true tags by the 1990s model and the 1830s model. We compare this for progressive sentences and random sentences. In all cases, both models were given the masked version of the sentences.

Figure 4 compares the predictions of the 1990s model to those of the 1830s model for the masked sentences. It shows that the 1990s model predicts the tags of the progressive sentences better than the 1830s model does, and the gap between their predictions is larger than it is for random sentences. This is evidence that the 1830s model is unfamiliar with progressive constructions.

## 7 Related Work

It has been claimed that the performance of Transformers on downstream tasks is insensitive to word order (Hessel and Schofield, 2021; Sinha et al., 2021a; Sinha et al., 2021b; Pham et al., 2021;

Gupta et al., 2021), which is consistent with our observation that BERT fine-tuned on shuffled text can distinguish between two time periods with high accuracy; however, our experiments indicate BERT's accuracy still improves when it has access to word order, consistent with Papadimitriou et al. (2022), who argue that word order sometimes matters by considering non-prototypical cases where meanings and lexical expectations do not match.

A sizable literature considers syntactic probes for BERT embeddings (e.g. Tenney et al., 2019; Jawahar et al., 2019; Hewitt and Manning, 2019; Pimentel et al., 2020), but to our knowledge none of the works that probe syntax consider Transformers fine-tuned on different time periods.

Syntactic change detection has been understudied in the computational linguistics and NLP literature. One work on this topic is Merrill et al. (2019), in which an LSTM achieved higher accuracy compared to a model consisting of feedforward networks on the task of predicting the year of composition of sentences. However, in addition to predicting the time period of text, we also identify specific instances of syntactic change. Computational methods for the study of syntactic change also appear in Hou and Smith (2018); this work examined possible causes of the decline in the rate of passivization over time. Another related work is Lei and Wen (2020), which found dependency distance to be decreasing over time (between the years 1790 and 2017).

Many studies have used BERT to detect semantic change, including Giulianelli et al. (2020), Hu et al. (2019), and Fonteyn (2020). In addition, Beelen et al. (2021) fine-tune historical BERT models on a corpus of 19th-century English books in order to tackle the task of time-sensitive Targeted Sense Disambiguation and find that "BERT works better when it is fine-tuned on data contemporaneous to the target period".

## 8 Conclusion

We investigated the extent to which fine-tuned BERT models can distinguish between the syntax of 1830s English and 1990s English. Our approach used masking techniques, and we repeatedly compared BERT models on masked and unmasked text.

First, in §2, we hid semantic information by replacing each word with its POS tag (and representing that tag as a common English word that exemplifies that tag). We fine-tuned BERT on such

syntax-only data to see if it could tell apart text from the 1830s and text from the 1990s. We found evidence that BERT can be used to detect syntactic differences between these two time periods.

In the later sections, we masked out individual words to see whether a POS tagger based on BERT embeddings (fine-tuned on either 1830s or 1990s) can still tag correctly when the word is masked. We filtered for a specific type of error, where the 1830s model predicts the correct tag when the word is masked but not when it is unmasked, and where the 1990s model is correct even when the word is unmasked. This procedure identified candidates for syntactic change, examples of which we provide in Appendix B. We were also able to confirm the rise of the progressive construction.

An important point to highlight is that our identification of syntactic change relied on an automatic POS tagger only for the 1990s text; we avoided running any automatic tagging software on older English data. In fact, from §3 onwards, all we did with the older 1830s data was to fine-tune BERT on it in an unsupervised fashion. Despite this restriction, we were able to automatically identify words whose part of speech had undergone change between the two time periods.

## Limitations

There are several limitations to this work. First, we worked only with English and only with text from the 1830s and 1990s. Our work also required the use of GPUs; see Appendix A for details of resources used. Although we were able to automatically identify specific instances of change (and we show automatically generated examples in Appendix B), our evaluation was limited to the manual judgement of one author, and we were unable to determine the recall of our detection system (because of the lack of objective criteria for what constitutes syntactic change). Additionally, our techniques are limited in the kinds of change we can detect: we focus on finding words whose typical part of speech has changed and on validating known syntactic changes such as the rise of the progressive. We hope that similar techniques can be extended to other types of syntactic change in future work.

## Acknowledgments

We thank Jacob Eisenstein and the anonymous reviewers for their helpful feedback.

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

## A  Resources Used

In this work, we mentioned and used the following licensed resources: BERT (Devlin et al., 2019), which uses Apache 2.0 license; UK Parliament Hansard Corpus (Alexander and Davies, 2015), which uses Open Parliament Licence; Trankit large version 1.1.0 (Nguyen et al., 2021), which uses Apache 2.0 license; and Stanford CoreNLP version 4.2.2 (Manning et al., 2014), which uses the GNU General Public License v3.0. All of these licenses allow the type of research conducted in our work. Moreover, our work falls within the intended use of each of these resources.

Our work required GPU computation. We used NVIDIA Tesla V100 and P100 GPUs over several months while working on the results of this paper. Running Trankit was the slowest step and took around 72 GPU hours; the total GPU time to reproduce our final results would be under a week.

## B  Automatically Identified Words

Table 2 shows the top of the list of automatically identified words whose part-of-speech usage may have changed; a longer list can be found in Appendix A of the first author's thesis (Hou, 2022). For each word, we give an example sentence from the 1990s on which the 1830s model was "confused". For each example sentence, we give the Trankit tag and two sets of probabilistic tag predictions given by the 1830s model; one set corresponds to the tags predicted by the 1830s model for the highlighted word when the word was masked, and the other set corresponds to the tags predicted by the 1830s model for the highlighted word when the word was unmasked. (For brevity, we only list the tag predictions to which the model assigns at least 30% probability.) An arrow such as 'JJ→Noun' indicates that the correct tag in the 1990s was 'Noun' but that the unmasked 1830s model put a high weight on 'JJ' instead; this suggests the word changed from 'JJ' in the 1830s to 'Noun' in the 1990s. (Recall that all POS tags for nouns were merged into a single 'Noun' category.)

| opposite JJ→Noun | "The mandatory life sentence does the **opposite** of what it sets out to achieve , and it should go ." | | |
| --- | --- | --- | --- |
| | Trankit: Noun | masked tag: 84% Noun | unmasked tag: 93% JJ |
| presume VB→VBP | "I **presume** that that shows they will be on message until they die ." | | |
| | Trankit: VBP | masked tag: 98% VBP | unmasked tag: 38% VB, 32% VBP |
| corporal Noun→JJ | "The school board wrote to me on 29 October advocating the reintroduction of **corporal** punishment in schools as a deterrent of last resort ." | | |
| | Trankit: JJ | masked tag: 74% JJ | unmasked tag: 95% Noun |
| intrepid JJ→Noun | "The second category are the Royal Marine s principal transport and landing ships , Fearless and **Intrepid** ." | | |
| | Trankit: Noun | masked tag: 64% Noun | unmasked tag: 97% JJ |
| evil Noun→JJ | "The gunmen , in the **evil** atrocity that they committed , symbolised the past ." | | |
| | Trankit: JJ | masked tag: 82% JJ | unmasked tag: 99% Noun |
| remembered JJ→VBN | "It should be **remembered** that , during the period 1974 – 79 , when he was a senior Cabinet Minister , the steel industry was inefficient and unprofitable ." | | |
| | Trankit: VBN | masked tag: 86% VBN | unmasked tag: 64% JJ, 36% VBN |
| shut VBD→VBN | "My second charge is that the Prime Minister has consistently talked about the incentive society with open opportunities for all , yet her policies have effectively **shut** out a quarter of the nation ." | | |
| | Trankit: VBN | masked tag: 97% VBN | unmasked tag: 86% VBD |
| offending JJ→Noun | "I can vouch for the fact that juvenile **offending** is a major problem in my constituency ." | | |
| | Trankit: Noun | masked tag: 98% Noun | unmasked tag: 58% JJ |
| granting VBG→Noun | "In most normal cases there will be the usual opportunity for consultation and **granting** of opinion from the local community ." | | |
| | Trankit: Noun | masked tag: 99% Noun | unmasked tag: 98% VBG |
| disposed Noun→VBN | "Mr. Streeter : To ask the Secretary of State for Defence what Ministry of Defence land and buildings in Plymouth he expects to be surplus to MOD requirement in the next five years and to be **disposed** of to the private sector ." | | |
| | Trankit: VBN | masked tag: 87% VBN | unmasked tag: 48% Noun, 31% VBN |
| exciting VBG→JJ | "We should also encourage housing associations to feel the **exciting** and exhilarating wind of competition ." | | |
| | Trankit: JJ | masked tag: 99% JJ | unmasked tag: 94% VBG |
| laying VBG→Noun | "More significant orders , especially those exercising the so - called Henry VIII powers , are to be subject to **laying** in draft and affirmative resolution ." | | |
| | Trankit: Noun | masked tag: 98% Noun | unmasked tag: 98% VBG |
| notwithstanding VBG→IN | "**Notwithstanding** signal failures and operational difficulties , I look forward to using that fast service ." | | |
| | Trankit: IN | masked tag: 84% IN | unmasked tag: 86% VBG |

Table 2: Automatically identified words whose part-of-speech usage may have changed