# OpenReview forum: "Detecting Syntactic Change with Pre-trained Transformer Models"
_EMNLP/2023/Conference — EMNLP 2023 Findings_

### Official Review · Reviewer_kxVY · 2023-07-31

**Soundness:** 3

**Excitement:**

3: Ambivalent: It has merits (e.g., it reports state-of-the-art results, the idea is nice), but there are key weaknesses (e.g., it describes incremental work), and it can significantly benefit from another round of revision. However, I won't object to accepting it if my co-reviewers champion it.

**Paper Topic And Main Contributions:**

This paper proposes a new way to detect syntactic changes between English texts from 1830 and the 1900s using the BERT model.
The paper has several contributions;
- The authors show that a fine-tuned version of BERT can distinguish texts from different periods using only syntactic information by performing different transformations, which validates the need for the next contributions
- They use a fine-tuned version of BERT to identify instances and words that are subject to syntactic change over time.
- They validate the historical rise of progressive construction using their method

The authors conducted a series of automatic and manual evaluations which is highly welcomed.

**Questions For The Authors:**

- Will the code and data be released? I did not see any mention of it.
- Section 3 is a bit confusing. The section starts with a paragraph saying that you train POS taggers, then later on we switch to classifiers. Should it be sequence taggers throughout the paper? Then we get into the training of MLM, and finally, the classification task using the learned representation. If the authors could make this section a bit clearer, we wouldn't need to read it over and over to wrap our head around what exactly is going on.
- Section 5.3: The authors expose the results from this experiment but nothing is said. What do these results tell us?

**Reasons To Accept:**

- The paper is straightforward and pretty easy to follow.
- The problem is well-motivated since they illustrate with clear experiments that Transformer-based models can identify syntactic change.
- The experiment was conducted both at the word *and* sentence level and makes the paper really interesting from an anecdotal perspective
- Section 6 is really interesting and I was left wanting more (see reasons to Reject)

**Reasons To Reject:**

- While the experiments are clear, well conducted and all sound, I feel like it is missing a concrete NLP application (especially for a NLP application track). See "questions for the authors" section.
- The manual evaluation lacks details and should be further expanded so that we can easily replicate the authors' experiment. Also, it has been mentioned in the limitations so not really impacting here, but it is at a really small scale.

**Reproducibility:**

4: Could mostly reproduce the results, but there may be some variation because of sample variance or minor variations in their interpretation of the protocol or method.

**Reviewer Confidence:**

3: Pretty sure, but there's a chance I missed something. Although I have a good feel for this area in general, I did not carefully check the paper's details, e.g., the math, experimental design, or novelty.

**Typos Grammar Style And Presentation Improvements:**

- Line 199: Is it from 1830 instead of the 1900s?
- Section 6: Would it be possible to have an example of progressive vs non-progressive construction? I think it'll improve this section, which in my opinion should be further emphasized.

---

> ### Author Rebuttal · Authors · 2023-08-29
>
> We appreciate the feedback.
>
> >*Will the code and data be released?*
>
> The data is publicly available from the UK government. We will release the code and the specific data splits that we used if the paper is accepted.
>
> >*The section starts with a paragraph saying that you train POS taggers, then later on we switch to classifiers. Should it be sequence taggers throughout the paper?*
>
> Our taggers are classifiers, as they take as input an embedding of a word and classify this embedding as one of 21 possible classes. Our taggers are not sequence-to-sequence taggers; instead, we use a BERT model to embed each word in a sentence, and we train a separate classifier to tag each word using only its embedding. (We do this twice, for two different BERT models, each fine-tuned on one decade.)
>
> >*Section 5.3: The authors expose the results from this experiment but nothing is said. What do these results tell us?*
>
> Of the top 100 candidate sentences that were automatically generated, our manual evaluation concluded that 33 of them were valid instances of syntactic change. We view this as a positive result, since a large fraction of the candidate changes were real, but one of our difficulties is that there is no literature on this task to which to compare our results.

---

### Official Review · Reviewer_wgbS · 2023-08-04

**Soundness:** 4

**Excitement:**

4: Strong: This paper deepens the understanding of some phenomenon or lowers the barriers to an existing research direction.

**Paper Topic And Main Contributions:**

This submission proposes to use fine-tuned language models to detect syntactic change in language.  They use transcriptions from the UK parliament from the 1830s and 1990s, automatically annotate them with POS tags, train a language model to predict POS tags (one for each epoch) and then check whether the two models disagree on their predictions.  The second kind of test is to mask a word, predict the words POS and check whether the model gets the POS right with masking but wrong with the word present. (Plus some more aspects that I leave out here)

In their first experiment, the authors try to remove confounders by replacing words with other words of the same POS in the training and test data and show that they can still train a model that can distinguish between new and old text; meaning that the LLM can distinguish the different POS sequences and does not rely on word identity. By also training on shuffled data, they show that the model is not only learning bag-of-word information but takes into account the actual sequence.  This experiment is convincing.

In the second and main experiment (which I described in the first paragraph), two models are being trained. I am confused by the reporting of accuracy, given that the models are not trained on some manually labeled ground truth but the output of another part of speech tagger. Additionally, the (necessary) inclusion of a tagger to create the training material may introduce a bias, as this tagger is probably trained on modern data -- I am not sure how to think about a tagger (the BERT model) that was trained on old text, which in turn was annotated by a tagger trained on modern text.

The authors then go on and find words in the 1990 text where
 - the 1830 model predicts an incorrect tag and
 - the 1830 model predicts a correct tag when the target word is masked and
 - the 1990 model predicts a correct tag.

This approach seems sound to me for finding a change, but to better understand to what extend we are dealing with noise and not shift, it would probably also have been a good idea to cross-check the instances where the 1990 model is wrong on 1990 text but the 1830 model is right.

I am not sure that all the experiments show that they extract syntactic shift. For example:
 - corporal (adj vs noun) might be due to a distribution shift of both meanings, exacerbated by the old adjective writing "corporeal"
 - inteprid occurs often as a noun because there is now a ship called "HMS Inteprid". This ship was under discussion due to ship renewals(?) in the 1990 and, as this ship did not exist in the 1830s, there was no proper noun usage back then. I do not think that people who have no interested in the British navy would now consider inteprid to be a noun.

In the third experiment, the authors take the same kind of test to the sentence level and look at the probability for all correct POS for a given sentence, again with the modes described above.  In the fourth experiment, they check whether sentences with progressives and show that the 1830 model assigns more probability to the correct tag sequence when the present participle is masked, which is not true (to the same extend) for random masking and also not true for the 1990 model.

Overall, I found this to be a good paper with a clear sequence of questions and clear description of the methods and reasoning behind the methods.

**Questions For The Authors:**

A) Maybe you can say something about the tag-old-data-with-tagger-trained-on-new-data aspect?

**Reasons To Accept:**

A good sequence of experiments building on top of each other that seem to be sound to me and which could be adapted to other questions.

**Reasons To Reject:**

No real reasons

**Reproducibility:**

3: Could reproduce the results with some difficulty. The settings of parameters are underspecified or subjectively determined; the training/evaluation data are not widely available.

**Reviewer Confidence:**

3: Pretty sure, but there's a chance I missed something. Although I have a good feel for this area in general, I did not carefully check the paper's details, e.g., the math, experimental design, or novelty.

---

> ### Author Rebuttal · Authors · 2023-08-29
>
> Thank you for your comments.
>
> >*Maybe you can say something about the tag-old-data-with-tagger-trained-on-new-data aspect?*
>
> Except in Section 2 (line 87 to line 175), we do not tag old data with a tagger trained on new data. For the main part of the paper, we only ever tag data from the 1990s. Our trick is to train a tagger that uses an older (1830s) understanding of English and have it tag 1990s data; we can then see where it makes mistakes. That is, we train BERT in an unsupervised fashion on the 1830s text. We then run this model to embed text from the 1990s and use these embeddings to train a tagger (trained only on 1990s data, but using the representations that were informed by an 1830s "understanding" of English). We identify instances of change by looking at tagging disagreements between this tagger and an analogous one based on 1990s data only; the tagging disagreements are checked for the 1990s data only. This way, all tags are only ever generated for the 1990s, and no tags are associated with text from the 1830s (outside of the separate analysis in Section 2).

---

### Official Review · Reviewer_mmN7 · 2023-08-06

**Soundness:** 3

**Excitement:**

3: Ambivalent: It has merits (e.g., it reports state-of-the-art results, the idea is nice), but there are key weaknesses (e.g., it describes incremental work), and it can significantly benefit from another round of revision. However, I won't object to accepting it if my co-reviewers champion it.

**Paper Topic And Main Contributions:**

The paper aims at an unsupervised approach to syntactic change detection based on BERT. It starts from a sentence-level classifier for distinguishing texts from different time periods based on syntactic information (while hiding semantic information) to identify words or phrases that manifest a relevant syntactic change. To this end, the paper focuses, among other things, on the detection of words that undergo POS changes. The main idea is to look for differences in two POS taggers specifically fine-tuned for the texts of the two "periods" under consideration, and to test whether they produce differences in the tagging of some test sentences when it comes to cases where a test word is masked once and then not: here, differences are taken as indicators of context-specific POS changes if the complementary model tends to not indicate such a difference.

**Questions For The Authors:**

1. I assume that the words used to replace the POS in section 2 are always the same for both periods and there is no effect of selecting different such words - correct?
2. I assume that the tags used in Trankit are not the same as those used in Stanford CoreNLP - but which ones?

**Reasons To Accept:**

1. The paper promises to avoid manual tagging of data to a larger degree, thus contributing to efficient modeling, especially in the area of historical texts where manual tagging is laborious and requires hard-to-find experts.
2. The paper proposes a simple but nice classifier to separate semantic and syntactic information, at least superficially.
3. The study of syntactic changes is an interesting area of research, especially from the point of view of using BERT and related approaches when semantic information is more or less ignored. The paper is an interesting proposal in this respect.

**Reasons To Reject:**

1. The paper considers a particular type of morpho-syntactic change as an example of syntactic change, by studying POS tagging changes in context as an indicator of system-level syntactic change, using examples of words that are likely (or at least to an unknown degree) to be subject to conversion (https://en.wikipedia.org/wiki/Conversion_(word_formation)) - e.g., sovereign. It is not so clear to me why it should be too hard to detect such "POS changes", or changes in preferences for using the word in either POS (e.g. noun vs. adjective): wouldn't something like a simple regular expression approach, working on parse trees or something, and using available lexicon information (https://en.wiktionary.org/wiki/sovereign), do much of the job? At least this information could be used to test the algorithm of the paper. In any case, conversions and similar examples of POS changes are not, in my opinion, such a good example of syntactic changes. Also, the database is quite small: only two "periods" of two years are considered. So the results may not be very meaningful, since we don't know what happened in the time in between. And how much do the results depend on the very specific genre of the underlying text base: can we really derive generalizations about syntactic changes in English from this? There are at least some doubts. From my point of view, the paper is more about the applicability of different fine-tuned models to texts from periods on which they were not trained, rather than providing a valid model of syntactic change. However, the paper does not systematically examine alternative models; there is no comparative evaluation. The test data are only taken for sentences from the 1990 period: why not for the other period - the underlying tests do not seem to be comprehensive enough. In addition, the model pretends not to be supervised. However, a single tagger for modern English is used as a reference (line 270), which means that information generated in a supervised manner is included in the model.
2. There is a kind of self-contradiction in the approach: the classification task requires the use of a POS tagger trained on standard contemporary English to historical text. I thought the authors wanted to highlight the problem of performing such an approach in line 071f.: "our method avoids running a POS tagger on old text" - and as admitted for Stanford CoreNLP by the authors in line 217f.; (I admit that from line 275 onwards the paper returns to its premise).
3. The classifier for separating semantic and syntactic information is nice, but one cannot tell from its use what syntactic information is actually represented. BERT represents some syntactic information, but what exactly: one does not learn that much about it. In other words, the supposed syntactic classifier is a black box, which hides the actual syntactic information in use. Is it just n-grams of certain words of certain POS in a certain sentence window? Or something more interesting in terms of syntax?
4. The model is not well described in all details: in particular, the part in lines 177-199 where it seems necessary to switch between the 1830- and 1990-related representations in order to transfer information from one "period" to the other is not understandable in the way it is described: I suspect that there is a simple algorithm behind this, but the description does not live up to this simplicity. Figure 1 is a big step forward in terms of comprehensibility, but the text itself falls short.

**Reproducibility:**

3: Could reproduce the results with some difficulty. The settings of parameters are underspecified or subjectively determined; the training/evaluation data are not widely available.

**Reviewer Confidence:**

3: Pretty sure, but there's a chance I missed something. Although I have a good feel for this area in general, I did not carefully check the paper's details, e.g., the math, experimental design, or novelty.

**Typos Grammar Style And Presentation Improvements:**

- "an automatic part-of-speech" -> "an part-of-speech" ("automatic tagger" is an pleonasm)
- "BERT is aware of syntactic differences" -> "BERT can be used to detect syntactic differences" (to speak of awareness is an anthropomorphism) - this should be corrected throughout thew paper
- "displays a poorer understanding of the progressive construction" -> "is a worse model for representing progressive construction"
- "we want to restrict to syntactic information" -> "we want to limit ourselves to syntactic information"
- "explosion.ai/blog/ud-benchmarks-v3-2" -> "https://trankit.readthedocs.io/en/latest/performance.html"
- "the model trained on text from the 1990s believes the word “co-operative” is an adjective" -> "the model trained on text from the 1990s tags the word “co-operative” as an adjective"

---

> ### Author Rebuttal · Authors · 2023-08-29
>
> We thank the reviewer for giving us feedback. The reviewer raised concerns about our approach, which we address here:
>
> >*the model pretends not to be supervised. However, a single tagger for modern English is used as a reference (line 270), which means that information generated in a supervised manner is included in the model*
>
> Our goal is to avoid supervision on the older English text. We allow supervision for text from the 1990s, and we never pretended otherwise. The reason to avoid using a POS tagger on English from the 1830s is that we do not have a trained tagger for this time period (nor do we have supervised training data for the 1830s).
>
> >*There is a kind of self-contradiction in the approach: the classification task requires the use of a POS tagger trained on standard contemporary English to historical text. I thought the authors wanted to highlight the problem of performing such an approach in line 071f.: "our method avoids running a POS tagger on old text" - and as admitted for Stanford CoreNLP by the authors in line 217f.; (I admit that from line 275 onwards the paper returns to its premise).*
>
> This only applies to Section 2, which is a self-contained section with a separate approach and a separate goal. The rest of the paper (including the main model) does not use such a POS tagger on historical text and does not use any classifier from Section 2. The reviewer mentions "line 275", but Section 2 is from line 87 to line 175 (not 275).
>
> >*The model is not well described in all details: in particular, the part in lines 177-199 where it seems necessary to switch between the 1830- and 1990-related representations in order to transfer information from one "period" to the other is not understandable in the way it is described*
>
> It is not necessary to switch between the 1830s and 1990s representations. We train two different instances of BERT for the two time periods, and we do not need to switch between the representations. We only compare the tagging predictions based on these BERT representations.
>
> We respond to the reviewer's questions below:
>
> >*1. I assume that the words used to replace the POS in section 2 are always the same for both periods and there is no effect of selecting different such words - correct?*
>
> The words are always the same for both periods. We did not test different word sets.
>
> >*2. I assume that the tags used in Trankit are not the same as those used in Stanford CoreNLP - but which ones?*
>
> Trankit and Stanford CoreNLP are both capable of outputting Penn Treebank Tags, so we deal with the same set of tags after switching taggers. Footnote #5 in our paper contains a link to the Penn Treebank.

---

### Meta-Review · Area_Chair_Cc5a · 2023-09-20

**Recommendation:** 3

**Metareview:**

The authors check the syntactic change in the UK parliament data from 1830 to 1990. They fine-tune a PLM to detect the syntactic change and also train their own POS tagger, which they compare to the fine-tuned PLM. The main contribution of the paper uses a hybrid tagger, a tagger that uses embeddings that have been shown to understand old English.

---

### Decision · Program_Chairs · 2023-10-07

**Decision:**

Accept-Findings

**Comment:**

The authors check the syntactic change in the UK parliament data from 1830 to 1990. They fine-tune a PLM to detect the syntactic change and also train their own POS tagger, which they compare to the fine-tuned PLM. The main contribution of the paper uses a hybrid tagger, a tagger that uses embeddings that have been shown to understand old English.